# Herpes zoster vaccine safety in the Aotearoa New Zealand population: a self-controlled case series study

James F. Mbinta [1], Alex X. Wang[2], Binh P. Nguyen [2], Janine Paynter [3], Prosper Mandela A. Awuni [4], Russell Pine [1], Andrew A. Sporle [5], Steve Bowe [1] & Colin R. Simpson [1,6]

In Aotearoa New Zealand, zoster vaccine live is used for the prevention of zoster and associated complications in adults. This study assessed the risk of pre-specified serious adverse events following zoster vaccine live immunisation among adults in routine clinical practice. We conducted a self-controlled case series study using routinely collected national data. We compared the incidence of serious adverse events during the at-risk period with the control period. Rate ratios were estimated using Conditional Poisson regression models. Falsification outcomes analyses were used to evaluate biases in our study population. From April 2018 to July 2021, 278,375 received the vaccine. The rate ratio of serious adverse events following immunisation was 0·43 (95% confidence interval [CI]: 0·37−0·50). There was no significant increase in the risk of cerebrovascular accidents, acute myocardial infarction, acute pericarditis, acute myocarditis, and Ramsay−Hunt Syndrome. The herpes zoster vaccine is safe in adults in Aotearoa New Zealand.

Varicella zoster virus causes two distinct diseases, varicella (primary varicella-zoster virus infection) occurring predominantly in children[1], and herpes zoster, commonly known as shingles, which occurs following reactivation of the latent varicella zoster virus[2].

In Aotearoa New Zealand, the zoster vaccine live, a single dose live attenuated vaccine, was licenced on 1st April 2018 for use in adults ≥ 50 years and funded for those 65 years old (catchup cohort: 66−80 years old, up to 31st December 2021)[3]. The zoster vaccine lives to boost cell-mediated immunity and significantly reduces the risk and severity of herpes zoster and associated complications[4]. The efficacy and effectiveness of the zoster vaccine live against herpes zoster and associated complications have been demonstrated in meta-analyses of randomised control trials and observational studies[5−11].

The safety of the herpes zoster vaccine has been demonstrated in prelicensure clinical trials and post-licensure observational studies[12], including inoculation site and systemic adverse events[8]. The relationship between serious adverse events (cardiovascular and cerebrovascular diseases, meningitis, encephalopathy, severe autoimmune diseases and disseminated varicella-zoster virus infection) has been investigated in post-licensure observational studies[13−16]. However, none of these studies have assessed the risk of serious adverse events (requiring hospitalisation) using whole country electronic health record data. Only two studies have investigated the safety of the zoster vaccine live in people with immunosuppressive conditions (haematologic malignancy and haematopoietic cell transplant)[17], and immunosuppressant drugs[18]. Also, to our knowledge, the safety of the zoster vaccine live has not been evaluated in Aotearoa New Zealand.

We assessed the risk of pre-specified serious adverse events among Aotearoa New Zealand adults who received the zoster vaccine live between 1st April 2018 and 24th July 2021. Post-licensure national

[1]School of Health, Wellington Faculty of Health, Victoria University of Wellington, Wellington, New Zealand. [2]School of Mathematics and Statistics, Wellington Faculty of Engineering, Victoria University of Wellington, Wellington, New Zealand. [3]Department of General Practice & Primary Healthcare, University of Auckland, Auckland, New Zealand. [4]School of Kinesiology and Health Sciences, Laurentian University, Sudbury, Canada. [5]iNZight Analytics Ltd., Department of Statistics, Faculty of Science, University of Auckland, Auckland, New Zealand. [6]Usher Institute, The University of Edinburgh, Edinburgh, UK. ✉e-mail: james.mbinta@vuw.ac.nz; colin.simpson@vuw.ac.nz

## Table 1 | Baseline characteristics of the study population

| Characteristics | Total N = 1082 | At risk N = 321 (29.67%) | Control N = 761 (70.33%) |
|---|---|---|---|
| **Year of hospitalisation** | | | |
| 2018 | 658 (60.81%) | 214 (66.67%) | 444 (58.34%) |
| 2019 | 237 (21.90%) | 65 (20.25%) | 172 (22.60%) |
| 2020 | 117 (10.81%) | 28 (8.72%) | 89 (11.70%) |
| 2021 | 70 (6.47%) | 14 (4.36%) | 56 (7.36%) |
| **Age group** | | | |
| 50–64 | ≤5 (0.27%) | ≤5 (0.31%) | ≤5 (0.26%) |
| 65–69 | 270 (24.95%) | 80 (24.92%) | 190 (24.97%) |
| 70–74 | 321 (29.67%) | 103 (32.09%) | 218 (28.65%) |
| 75–76 | 376 (34.75%) | 109 (33.96%) | 267 (35.09%) |
| ≥80 | 112 (10.35%) | 28 (8.72%) | 84 (11.04%) |
| **Sex** | | | |
| Male | 642 (59.33%) | 187 (58.26%) | 455 (59.79%) |
| Female | 440 (40.67%) | 134 (41.74%) | 306 (40.21%) |
| **Ethnicity (Level 1 ethnic codes)** | | | |
| European (1) | 779 (72.00%) | 222 (69.16%) | 557 (73.19%) |
| Māori (2) | 147 (13.59%) | 51 (15.89%) | 96 (12.61%) |
| Pacific Peoples (3) | 65 (6.01%) | 18 (5.61%) | 47 (6.18%) |
| Asian (4) | 79 (7.30%) | 25 (7.79%) | 54 (7.10%) |
| Others (5,6,9) | 12 (1.11%) | ≤5 (1.55%) | 7 (0.92%) |
| **New Zealand Index of Deprivation 2013** | | | |
| Quintile 1 | 171 (15.80%) | 47 (14.64%) | 124 (16.29%) |
| Quintile 2 | 170 (15.71%) | 45 (14.02%) | 125 (16.43%) |
| Quintile 3 | 216 (19.96%) | 69 (21.50%) | 147 (19.32%) |
| Quintile 4 | 242 (22.37%) | 73 (22.74%) | 169 (22.21%) |
| Quintile 5 | 283 (26.16%) | 87 (27.10%) | 196 (25.76%) |
| **New Zealand Index of Deprivation 2018** | | | |
| Quintile 1 | 162 (14.97%) | 45 (14.02%) | 117 (15.37%) |
| Quintile 2 | 191 (17.65%) | 53 (16.51%) | 138 (18.13%) |
| Quintile 3 | 225 (20.79%) | 68 (21.18%) | 157 (20.63%) |
| Quintile 4 | 239 (22.09%) | 71 (22.12%) | 168 (22.08%) |
| Quintile 5 | 265 (24.49%) | 84 (26.17%) | 181 (23.78%) |
| **Immune suppression** | | | |
| Yes | 93 (8.60%) | 30 (9.35%) | 63 (8.28%) |
| No | 989 (91.40%) | 291 (90.65%) | 698 (91.72%) |
| **District Health Board** | | | |
| Auckland | 79 (7.30%) | 26 (8.10%) | 53 (6.96%) |
| Bay of Plenty | 57 (5.27% | 17 (5.30%) | 40 (5.26%) |
| Canterbury | 129 (11.92%) | 43 (13.40%) | 86 (11.30%) |
| Capital and Coast | 49 (4.53%) | 12 (3.74%) | 37 (4.86%) |
| Counties Manukau | 116 (10.72%) | 38 (11.84%) | 78 (10.25%) |
| Hawke's Bay | 34 (3.14%) | 10 (3.12%) | 24 (3.15%) |
| Hutt Valley | 42 (3.88%) | 12 (3.74%) | 30 (3.94%) |
| Lakes | 28 (2.59%) | ≤5 (1.56%) | 23 (3.02%) |
| MidCentral | 58 (5.36%) | 19 (5.92%) | 39 (5.12%) |
| Nelson Marlborough | 30 (2.77%) | 9 (2.80%) | 21 (2.76%) |
| Northland | 57 (5.27%) | 17 (5.30%) | 40 (5.26%) |
| South Canterbury | 16 (1.48%) | ≤5 (0.62%) | 14 (1.84%) |
| Southern | 83 (7.67%) | 23 (7.17%) | 60 (7.88%) |
| Tairawhiti | 8 (0.74%) | 0 (0.00%) | 8 (1.05%) |
| Taranaki | 17 (1.57%) | ≤5 (1.56%) | 12 (1.58%) |
| Waikato | 109 (10.07% | 27 (8.41%) | 82 (10.78%) |
| Wairarapa | 11 (1.02%) | ≤5 (0.62%) | 9 (1.18%) |

## Table 1 (continued) | Baseline characteristics of the study population

| Characteristics | Total N = 1082 | At risk N = 321 (29.67%) | Control N = 761 (70.33%) |
|---|---|---|---|
| Waitemata | 120 (11.09%) | 42 (13.08%) | 78 (10.25%) |
| West Coast | 13 (1.20%) | 6 (1.87%) | 7 (0.92%) |
| Whanganui | 26 (2.40%) | 6 (1.87%) | 20 (2.63%) |

New Zealand Index of Deprivation (2013 and 2018) measures relative socioeconomic deprivation. It combines data related to communication, income, employment, qualifications, house ownership, support, living space and living condition. It is divided into quintiles (Quintile 1 = least deprived; Quintile 5 = most deprived), with each quintile representing 20% of the population.
At risk: Vaccine recipients with adverse events within 1–42 days post-vaccination.
Control: Vaccine recipients with adverse events within 73–162 days post-vaccination.

observational studies offer a unique opportunity to assess rare adverse outcomes requiring a large sample size.

## Results

### Baseline characteristics of the study population

From April 2018 to July 2021, 278,375 adults were vaccinated with the zoster vaccine live. The mean age was 71.1 (standard deviation = 5.0). Of these, 161,217 (58.9%) were ≥70 years old, 145,360 (52.2%) were female, 225,707 (81.1%) were European, and 16,964 (6.1%) had immunocompromising conditions. During the study period, 16,081 (5.8%) Māori and 9170 (3.3%) Pacific people who received the zoster vaccine lived.

A total of 1082 (0.4%) vaccinated people were hospitalised for a pre-specified adverse event of interest during the at-risk (n = 321) and control (n = 761) periods and were included in the study population. The mean age of the study population with an adverse event was 73.4 (standard deviation = 5.0; range: 58–92), and 809 (74.8%) were ≥70 years old. Furthermore, 59.3% (n = 642) of these people were males, 72.0% (n = 779) were European, and 8.6% (n = 93) were immunocompromised (Table 1). Most of the study participants (24.5%) were from areas with the most deprived scores.

### Pre-specified adverse events

Overall, 1431 serious adverse events occurred during the study period (Fig. 1 and Table S6). After assessment against the inclusion criteria, 1130 pre-specified adverse events were identified during 124,741 person-days of follow-up. There were 320 and 810 serious adverse events in the at-risk (40,775 person-days) and control (64,204 person-days) periods, respectively. The rate ratio of serious adverse events following immunisation was 0.43 (0.37–0.50) (Table 2).

There were 488 cases with a group one adverse event (cerebrovascular diseases) identified during the study period. The rate ratio was 0.46 (95% CI: 0.36–0.56). There was no increase in the risk of stroke (haemorrhage or infarction) during the at-risk period (134 events during 17,677 person days) compared to the control period (352 cases during 28,642 person-days of follow-up): rate ratio = 0.45 (95% CI: 0.36–0.56). The rate ratio for transient cerebral ischaemia was 2.07 (95% CI: 0.08–56.10).

Cardiovascular diseases were the most common pre-specified adverse events requiring hospitalisation (group two). Overall, 182 cardiovascular events occurred during 22,970 person-days of follow-up in the at-risk period compared to 454 events in 35,367 person-days (control period). The rate ratio was 0.41 (95% CI: 0.34–0.50). The rate ratios for acute myocardial infarction, acute pericarditis and acute myocarditis were 0.28 (95% CI: 0.16–0.47), 0.44 (95% CI: 0.12–1.33) and 0.21 (95% CI: 0.01–2.12), respectively.

No group three adverse events (meningitis, encephalitis and encephalopathy) were identified during the entire follow-up period, and the rate ratio was therefore not estimated.

There were six serious adverse events for group four (Ramsay–Hunt syndrome, Bell's palsy and facial palsy) which occurred during

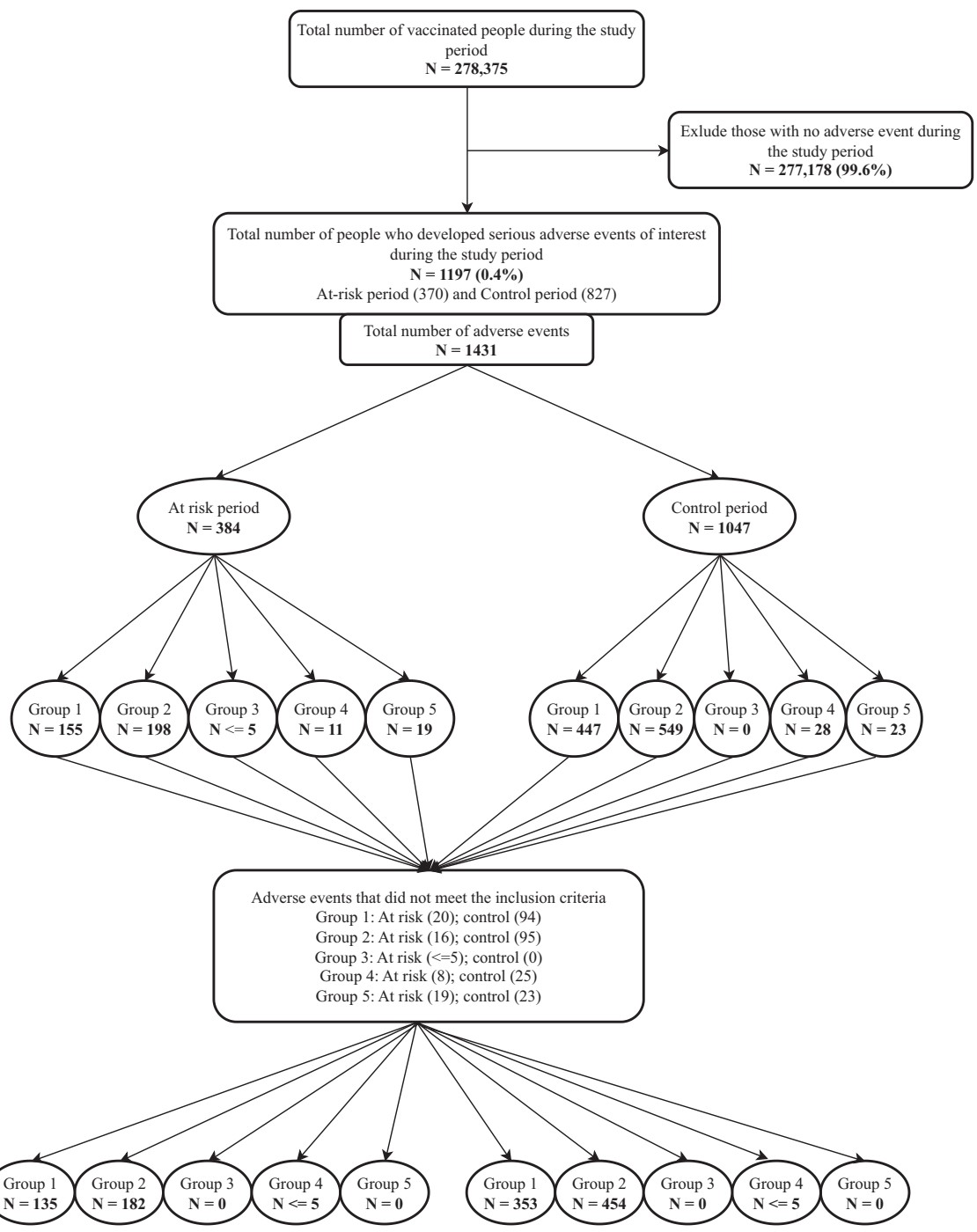

**Fig. 1 | Data flow chart.** A total of 278,375 adults were vaccinated during the study period (April 2018 to July 2021). A total of 1082 people developed pre-specified serious adverse events during the follow-up period and were included in the analysis. Group 1: Stroke, Cerebrovascular diseases; Group 2: Cardiovascular events; Group 3: Meningitis, encephalitis and encephalopathy; Group 4: Ramsay–Hunt Syndrome, Bell's Palsy; Group 5: Medically attended reactions.

the follow-up period (116 person-days). There was no difference in the risk of group four adverse events when comparing the at-risk and control periods (rate ratio = 0.79 (95% CI: 0.11–5.68).

There were no reported medically attended reactions (the group five serious adverse events–cellulitis and infections, and allergic reactions) during the follow-up period, and the rate ratio was not estimated.

### Falsification outcomes

The incidence of negative controls in the at-risk period was compared to the control window (Table 3). These negative controls should not

be associated with herpes zoster and zoster vaccine live vaccination (adverse events following immunisation). There was no difference in incidence between the at-risk and control periods for diverticulitis (rate ratio = 0.65; 0.08–3.78); femoral fractures (rate ratio = 1.51; 0.42–5.19); and cholecyst and pancreatic diseases (rate ratio = 4.38; 0.99–22.15).

## Discussion

In this self-controlled case series study using routinely collected national data, we found no increased risk of serious adverse events (requiring in-patient hospitalisation) within the at-risk period following zoster vaccine live vaccination in adults in Aotearoa New Zealand.

**Table 2 | Rate ratio of pre-specified adverse events following herpes zoster vaccination**

| Pre-specified adverse events | At risk period | | Control period | | Rate ratio (95% CI) |
|---|---|---|---|---|---|
| | Cases | PT | Cases | PT | |
| **Overall** | **320** | **40,775** | **810** | **64,204** | **0.43 (0.37–0.50)[a]** **0.42 (0.37–0.49)[b]** |
| **Group 1. Stroke, cerebrovascular diseases** *(First episode in 12 months)* | | | | | |
| Occlusion and stenosis | 0 | 0 | 0 | 0 | – |
| Transient cerebral ischaemia | ≤5 | 67 | ≤5 | 140 | 2.07 (0.08–56.10)[a] 2.07 (0.12–36.83)[b] |
| Acute, and other ill-defined, cerebrovascular disease | 0 | 0 | 0 | 0 | – |
| Haemorrhage or Infarction | 134 | 17,677 | 352 | 28,642 | 0.45 (0.36–0.56)[a] 0.45 (0.36–0.56)[b] |
| All group 1. adverse events | 135 | 17,744 | 353 | 28,782 | 0.46 (0.36–0.56)[a] 0.46 (0.37–0.57)[b] |
| **Group 2. Cardiovascular events** *(First episode in 12 months)* | | | | | |
| Acute myocardial infarction | 24 | 3481 | 73 | 5035 | 0.28 (0.16–0.47)[a] 0.28 (0.17–0.47)[b] |
| Acute pericarditis | ≤5 | 605 | 12 | 926 | 0.44 (0.12–1.33)[a] 0.44 (0.14–1.40)[b] |
| Acute myocarditis | ≤5 | 172 | ≤5 | 288 | 0.21 (0.01–2.12)[a] 0.21 (0.02–2.77)[b] |
| Cardiomyopathy | 6 | 1397 | 31 | 1880 | 0.14 (0.05–0.35)[a] 0.14 (0.05–0.37)[b] |
| Heart failure | 146 | 17,315 | 334 | 27,238 | 0.47 (0.37–0.58)[a] 0.47 (0.38–0.58)[b] |
| All group 2. adverse events | 182 | 22,970 | 454 | 35,367 | 0.41 (0.34–0.50)[a] 0.40 (0.33–0.49)[b] |
| **Group 3. Meningitis, encephalitis and encephalopathy** *(First episode in 12 months)* | | | | | |
| Meningitis, encephalitis and encephalopathy | 0 | 0 | 0 | 0 | – |
| **Group 4. Ramsay–Hunt syndrome, Bell's Palsy** *(First episode in 12 months)* | | | | | |
| Ramsay–Hunt Syndrome, Bell's Palsy | ≤5 | 61 | ≤5 | 55 | 0.79 (0.11–5.68)[a] 0.79 (0.11–5.48)[b] |
| **Group 5. Medically attended Reactions** *(First episode in 30 days)* | | | | | |
| Cellulitis and infection | 0 | 0 | 0 | 0 | 0 |
| Allergic reactions | 0 | 0 | 0 | 0 | 0 |

*PT* person-time (person-days).
[a]Used gnm to fit a conditional Poisson regression model.
[b]Used clogit to fit a conditional logistic regression model.

**Table 3 | Falsification outcomes (negative controls)**

| Outcomes | At risk | | Control | | Rate ratio (95% CI) |
|---|---|---|---|---|---|
| | Cases | PT | Cases | PT | |
| Diverticulitis | ≤5 | 215 | ≤5 | 352 | 0.65 (0.08–3.78) |
| Femoral fracture | 6 | 436 | 7 | 857 | 1.51 (0.42–5.19) |
| Cholecyst and pancreatic diseases | 6 | 291 | ≤5 | 682 | 4.38 (0.99–22.15) |

*PT* person-time (person-days).
We found no pertussis, hernia, renal calculi, epistaxis, haemorrhoids, appendicitis, burns and lipoma-related hospitalisations in the at-risk and control windows.

These results are consistent with previous pre- and post-licensure safety studies of the zoster vaccine live. In a network meta-analysis of randomised controlled trials involving 103,899 adults 50 years and older, Tricco et al. found no increased risk (risk ratio = 1.08; 95% CI: 0.91–1.30) of serious adverse events (requiring hospitalisation) within 30 days of herpes zoster vaccination[5]. An updated Cochrane review (including 24 randomised controlled trials and 88,551 adults ≥ 60 years) found no statistically significant difference in the risk of serious adverse events between the vaccinated and placebo groups

(risk ratio = 1.08; 95% CI: 0.95–1.21)[8]. A network meta-analysis of 21 randomised controlled trials by McGir et al., showed that there was no statistically significant difference in the risk of serious adverse events following vaccination between the vaccine and placebo groups for any formulation (subcutaneous. intramuscular and refrigerated) of the zoster vaccine live[6]. In a long-term follow-up study, the risk of serious adverse events, hospitalisation and death were similar in the vaccinated and placebo groups[12]. A 2019 GRADE analysis of 21 randomised controlled trials and seven observational studies confirmed that there were no severe adverse events associated with zoster vaccine live vaccination[19]. The rate of serious adverse events in the vaccinated and placebo groups was similar.

A cohort study involving 29,000 people ≥ 60 years who received the zoster vaccine live from July 2006 to November 2007 found no increased risk of serious adverse events (resulting in hospitalisation) within 42 days post-vaccination[14]. Serious pre-specified adverse events assessed using electronic diagnostic codes and review of medical records were coronary atherosclerosis and other heart diseases (relative risk = 1·79; 95% CI: 0.69–4.51), acute myocardial infarction (relative risk = 1.29; 95% CI: 0.66–2.43); and stroke (relative risk = 0.91; 95% CI: 0.43–1.81). Using a case-centred study and SCCS designs (electronic

diagnostic code and review of medical records), Tseng et al., examined a large cohort of 193,083 adults ≥ 50 years old for increased risk of pre-specified adverse events following zoster vaccine live vaccination[13]. In both study designs, they observed no difference in risk of serious adverse events between the at-risk and control windows: cerebrovascular events (relative risk = 0.99; 95 CI: 0.83–1.19); meningitis, encephalitis and encephalopathy (relative risk = 0.78; 95% CI: 0.39–1.56); Ramsey–Hunt syndromes and Bell's palsy (relative risk = 0.78; 95% CI: 0.29–2.09); and cardiovascular diseases (relative risk = 0.99; 95% CI: 0.86–1.14). In a recent SCCS study in Australia involving 150,054 adults 70–79 years old (vaccinated between November 2016 and July 2018), there was no increased risk of serious adverse events within 42 days following zoster vaccine live inoculation: stroke (relative incidence = 0.58; 95% CI: 0.44–0.78); myocardial infarction (relative incidence = 0.74; 95% CI: 0.41–1.33); and medical attendance (relative incidence = 0.94; 95% CI: 0.94–0.95). In the sensitivity analysis, burns (relative incidence = 1.23; 95% CI: 0.97–1.57) were used as a negative control to assess the risk of bias[16]. Active surveillance of 17,000 adults 70–79 years using a short message service (self-reported) found no increased risk of adverse events (rate ratio = 1.05; 95% CI: 0.93–1.18) and medical attendance (rate ratio = 0.98; 95% CI: 0.50–1.91) when zoster vaccine live was concomitantly administered with other vaccines[20]. In a case–control study using the Vaccine Adverse Event Reporting System, Lai and Yew found no significant association between autoimmune adverse events and zoster vaccine live vaccination except arthritis (odds ratio = 2·7; 95% CI: 1.7–4.3), and alopecia (odds ratio = 2.2; 95% CI: 1.2–4.3)[15].

Evidence from epidemiological studies has shown the risk of myocardial infarction and stroke, and transient ischaemic attack significantly increased in the first month following the reactivation of the varicella-zoster virus and decreased with time since infection. The risk of stroke is two to three times higher than the baseline level when reactivation occurs in the ophthalmic branch of the fifth cranial (trigeminal) nerve[21]. A recent systematic review and meta-analysis and a nationwide retrospective matched observational study found that the zoster vaccine live effectively prevented herpes zoster hospitalisation in people with cerebrovascular disease and heart disease[9,10]. In the current study, we identified a large (>50%) protective effect of the zoster vaccine live for most of the outcomes (stroke, acute myocardial infarction and heart failure) in the first 42 days following zoster vaccine live vaccination. A significantly reduced risk of stroke (relative incidence = 0.58; 95% CI: 0.44–0.78) following the zoster vaccine live immunisation has been reported in a recent self-controlled case series study[16]. The main difference between our findings and numerous antecedent studies is that measure of association for the pre-specified adverse events was closer to one, or the confidence interval of the measure crossed one[5,6,8,13,16,22]. The apparent protective effect observed in our study could be attributed to the zoster vaccine's live effectiveness against herpes zoster and associated complications, the healthy vaccine effect, delayed serious adverse events following vaccination, and healthy vaccine bias and healthy vaccinee time[14,16]. Although a previous study[9] comparing the vaccinated (from which the current sample was selected) and the unvaccinated population found that both cohorts were comparable, further investigations are required to explore the significantly reduced risk of these outcomes following zoster vaccination.

Using representative national data, we have provided evidence that the zoster vaccine live was safe in the older New Zealand population (including immunocompromised). The SCCS method was used to assess the risk of pre-specified adverse severe events requiring hospitalisation. This study design controlled for measurable but not measured and unmeasurable confounding factors (e.g. sex, ethnicity, body mass index, smoking, genetics, frailty, level of immunosuppression and severity of co-morbidities) since comparisons were within the same individuals (each case was its control)[23]. The strength of this

approach is that it minimises biases introduced when comparing vaccinated and unvaccinated cohorts and has, using nationally linked health record datasets, been previously used to assess the safety of COVID-19 vaccines[24–26]. We used a Conditional Poisson regression model with offset for the time at risk. We assumed that serious adverse events were independent, vaccine uptake was high, exposure was transient and the time at-risk post-vaccination was short[23]. Since events are independent, we were able to assess multiple outcomes with varying observation periods. In the falsification outcome analysis, multiple negative controls were used to assess bias in our study population[16]. In the selection process for negative controls, further consideration was made to try and identify controls which have a similar severity and prevalence to the pre-specified adverse events[27]. There was a possible borderline increase in the risk of cholecyst and pancreatic diseases. However, the confidence intervals were wide and crossed 1.0, so we are still satisfied that our study is not subject to bias. We used pre-existing Ministry of Health databases, and the data are considered accurate and complete. We excluded all adverse events registered on the index date except anaphylaxis, as these were likely to have begun prior to zoster vaccine live vaccination[13].

We assumed the registration date in the National Immunisation Registry was the date the zoster vaccine live was administered (index date). Pre-specified serious adverse events were identified using the ICD-10-AM-iii diagnostic codes in electronic medical records. The positive predictive value of ICD codes compared to the review of the medical records is ≥83%[28,29]. This may have led to some underestimation of outcomes and a spurious association between adverse events and zoster vaccine live vaccination[30]. Misclassification and miscoding of adverse events and co-morbidities are inherent limitations of observational studies that use electronic medical record data[16].

We also assumed the zoster vaccine live was administered independently. In Australia's MedicineInsight programme, 92,857 adults 70–79 years received the zoster vaccine live from November 2016 to July 2018. Zoster vaccine live was administered independently (82%) and concomitantly with inactivated influenza vaccine (16%) and 23-valent pneumococcal vaccine (2%). There was no significant difference in risk of adverse events when the zoster vaccine live was modelled independently or with other vaccines[16]. The serious adverse events observed during the study period may be associated with other vaccines.

In New Zealand, the zoster vaccine live is approved for use in adults ≥ 50 years and used as part of the vaccination programme at age 65 from April 2018 to August 2022. The current guidelines preferentially recommend the recombinant zoster vaccine for the prevention of herpes zoster and postherpetic neuralgia in adults (≥50 years old) and immunocompromised people (≥18 years old). Two doses of the recombinant zoster vaccine were funded for adults 65 years old[3].

In conclusion, we found that the zoster vaccine live vaccination was not associated with serious adverse events (requiring hospitalisation) in New Zealand adults (including people with immunocompromising conditions). Subsequent research should systematically assess all adverse events following immunisation by linking general practice to Ministry of Health databases. Comprehensive active surveillance should be used for monitoring participants post-vaccination to identify rare serious adverse events.

## Methods
### Ethics and permissions
An ethics exemption (21/NTB/118) was obtained from the Health and Disability Ethics Committee of the Ministry of Health, New Zealand. After review and discussion by the Health and Disability Ethics Committees, the reviewers were satisfied that despite the large datasets being linked, there would be no way of identifying individuals, and the risk was minimal.

## Study setting

We used routinely collected national data available at the Ministry of Health (Table S1). The National Health Index was used to link data extracted from multiple datasets. Clinical, discharge and demographic information of hospitalised patients (in-patients and day patients in public and private hospitals) were obtained from the national minimum datasets[31]. Detailed clinical and pathological data (morphology code, basis of diagnosis, laboratory code and extent of disease) about cancer and demographic information of patients with malignancy were extracted from the New Zealand cancer registrations[32]. The national immunisation register (NIR) is an accurate electronic medical record that contains immunisation information on childhood and adult vaccines administered in New Zealand[33]. Detailed vaccine-related information (vaccine, event ID, event status description, vaccination date, indication, description and vaccine dose) was extracted from the NIR. The Pharmaceutical Collection is an electric record of all pharmaceuticals funded by the government and dispensed in the community. We extracted information on antiviral drugs and immunosuppressive drugs (Table S2)[34]. These datasets are exhaustive and have been used for research, public health surveillance and policy formulation[31].

## Study population

In step one, we identified all individuals who had completed herpes zoster vaccination (a single dose intramuscular injection of 0.65 mL Zostavax®) between 1st April 2018 and 24th July 2021 and were born in 1971 or earlier.

In step two, we identified all individuals who had a publicly funded hospital discharge where the date of admission fell within 42 days (inclusive) of the herpes zoster vaccination date (index date), and there was any diagnosis of any of the pre-specified outcomes in Table S4. We considered the date of admission as the day of onset of the adverse event. For each master NHI number identified in step two, we eliminated all publicly funded hospital discharges with an admission date that falls within one year of the admission date of the index event (except for group five adverse events). This was to ensure that the adverse event identified was the first event in 12 months (except for group five, where we considered the first event in 30 days)[13].

For each master encrypted NHI number with a completed herpes zoster vaccination identified in step two, we extracted all publicly funded hospital discharges with an admission date that falls within 73 and 162 days (inclusive) of the vaccination date. For each master NHI number, we excluded all publicly funded hospital discharges with an admission date that fell within one year of the admission date of the index event (except group five adverse events). In scenarios where a person had more than one index event, each index event had its year of retrospective data included.

The final study population comprised individuals with the outcome of interest that occurred during the follow-up (predefined observation) period. The follow-up period is the time after vaccination when we monitored the vaccinated cohort for any pre-specified adverse events and was divided into the at-risk (1–42 days) and the control (73–162 days) periods. This study design has the advantage of controlling for all time–fixed individual-level confounding variables as comparisons are within the same individual rather than between vaccinated and unvaccinated cohorts[23,35,36]. For each outcome of interest, we noted the event ID, start date, clinical code and diagnosis type (primary or secondary diagnosis).

## Study design

A self-controlled case series (SCCS) study design was used to estimate the risk of pre-specified serious adverse outcomes following herpes zoster vaccination (Table S3)[23]. SCCS is bidirectional; hence the control window can be pre or post-vaccination[35]. The control interval (background level) and post-vaccination period (risk interval) within each person were derived for every serious adverse event (requiring hospitalisation) and used to classify an individual as unexposed or exposed[23]. We assessed multiple independent pre-specified serious adverse events in the same individuals and acute adverse outcomes that occurred immediately post-vaccination[37].

The at-risk period was a duration of 42 days (days 1–42) following receipt of the zoster vaccine live vaccination (index date) (Fig. 2). Pre-specified adverse events that occurred on the day of vaccination (index day; Day 0) were excluded from the analyses. The at-risk period of 42 days is consistent across pre- and post-licensure studies that assessed the safety of herpes zoster vaccines[13,14,16,38]. A 30-day washout interval (day 42 to day 72) was included to reduce the probability of spillover of vaccine-attributable risks into the control window[35]. The post-risk period (post-vaccination self-comparison window) was 90 days, beginning on day 73 following vaccination. The post-risk period was set at 90 days to increase the chance of capturing relevant adverse events. Also, 90 days of follow-up compensated for the absence of a pre-risk period. In a previous study conducted by Baxter et al. in 2012, the rate of all

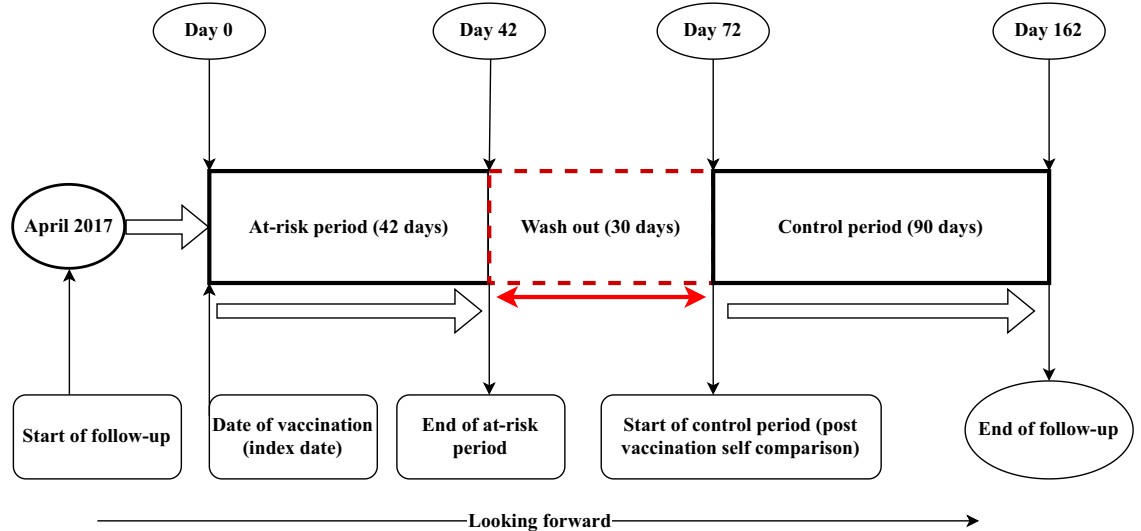

**Fig. 2 | Self-controlled case series study design for the analysis of pre-specified serious adverse events following zoster vaccine live vaccination.** Each vaccinated person served as their own control, hence eliminating the need for a separate control group. The follow-up period was divided into the at-risk and control (postvaccination comparison or baseline) periods. We introduced a washout period of 30 days to eliminate the carryover effects hence the outcomes during the control period were not influenced by the previous exposure (vaccination).

diagnoses of pre-specified adverse events of interest listed for hospitalisations and emergency department visits occurring in a 42-day risk period (days 1–42) following vaccination with herpes zoster live was compared to that in a 90-day postvaccination comparison period (days 91–180 post-vaccination), using vaccinees as their own controls[14]. Also, the control period of 90 days has been used in recent studies to assess adverse events associated with live attenuated and trivalent inactivated influenza vaccination[36] and to evaluate the associations between exposure to ChAdOx1 or BNT162b2 vaccination and haematological and vascular outcomes using a nested incident-matched case-control study and a confirmatory self-controlled case series analysis[25].

## Pre-specified adverse events

Serious adverse events associated with herpes zoster vaccination were defined using the International Statistical Classification of Diseases and Related Health Problems, Tenth Revision, Australian Modification (ICD-10-AM-iii)[39] and extracted from the electronic medical records (NMDS). These outcomes (Table S4) were divided into five main clinically related groups: Group one: Cerebrovascular diseases (occlusion and stenosis, transient cerebral ischaemia, acute and another ill-defined cerebrovascular disease, and haemorrhage or Infarction); Group two: Cardiovascular diseases (acute myocardial infarction, acute pericarditis, acute myocarditis, cardiomyopathy and heart failure); Group three: meningitis, encephalitis and encephalopathy, Group four: Ramsay–Hunt Syndrome and Bell's palsy (geniculate herpes zoster, herpetic geniculate ganglionitis, Bell's palsy and facial palsy) and Group five: medically attended reactions (cellulitis and infection and allergic reactions)[13]. The selection and classification of these serious adverse events were based on previous studies and biological plausibility[12–14,16,21].

## Falsification outcomes

In the self-controlled case series, only vaccinated people who developed the pre-specified adverse event during the follow-up period were included in the analysis. The follow-up period was divided into the at-risk window and the control (baseline) period. Since the duration of follow-up is short, baseline characteristics such as co-morbidity, sex, prioritised ethnicity and index of deprivation were self-controlled[40,41]. Differences might exist between the at-risk and control windows in terms of time-varying residual confounders. We assessed for bias in our study population by comparing the incidence of possible causes of hospitalisation (not previously associated with herpes zoster and related complications, i.e., negative controls) in the at-risk window to the control period[9,42,43]. These negative controls (appendicitis, hernia, diverticulitis, femoral fracture, cholelithiasis and cholecystitis, pancreatic diseases, sepsis, haemorrhoids, renal calculi, burns, lipoma and epistaxis) were extracted from the electronic database (NMDS) (Table S5).

## Statistical analysis

Proportions were used to describe the baseline characteristics of the study population. These baseline characteristics included age, sex, ethnicity, index of deprivation (Table S1), location (District Health Board) and immunosuppression (immune-compromising diseases and immunosuppressive therapy; Table S2). A single case contributed to exposed (at-risk period) and unexposed (control period) person-time. The incidence rate of an adverse event in the at-risk window (code = 1) was compared to that in the control interval (code = 0). The incidence in the at-risk window was estimated as the number of pre-specified adverse severe events occurring while individuals contributed to the respective at-risk period divided by the total person-days contributed to the at-risk period. The incidence in the control period was computed as the number of pre-specified adverse severe events occurring during the control period divided by the total person-days contributed to the control period (this was considered the baseline level).

Conditional Poisson and conditional logistic regression models were used to derive rate ratios and 95% CI for pre-specified adverse

events with offset for person-time. Rate ratios were estimated at the level of groups (clinically related adverse events) because of the small number of serious adverse events per diagnostic code. This aggregation of diagnostic codes has been used in previous studies[13]. The analyses were carried out by J.M. and A.W. and independently checked by B.N. and S.B. All statistical analyses were carried out using R/R Studio (version R-4.0.5) and Python (version 3.7.12)[44,45].

## Reporting summary

Further information on research design is available in the Nature Portfolio Reporting Summary linked to this article.

## Data availability

The data used in this study are based on de-identified national clinical records and will not be made publicly available. Access is controlled for privacy, confidentiality and legal issues (under the provisions of the New Zealand Privacy Act 2020, the New Zealand Health Information Privacy Code 2020 and any other legislation specifically referred to by the Ministry in the supply of this information). These are, however, available by application via the Ministry of Health New Zealand. Details can be obtained by contacting the Analytical Services team (data-enquiries@health.govt.nz), with the specification of the information required. The team will acknowledge the request and respond within 3–5 working days. It will take 3–4 weeks from receiving written confirmation to fulfil the request. The data agreement stipulates that the information provided is to be used for the purposes set out in the proposal/documentation provided to the New Zealand Ministry of Health (the Ministry) and is for your internal use only. It should not be made available to any other party without the Ministry's prior consent; this includes requests from academic journals for the supply of research datasets. The only exceptions to this would be the dissemination of data through research documents and/or statistical publications[46].

## Code availability

All code used in this study is publicly available at Vaccine-studies[47].

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

## Acknowledgements

A Wellington Doctoral Scholarship was awarded to J.F.M. The sponsors of the study had no role in the study design, data collection, data analysis, data interpretation or the writing of this report. There was no input to the methodology and results of this study by any commercial entity.

## Author contributions

J.F.M., B.P.N., J.P. and C.R.S. conceived the idea of the study and developed the protocol. All authors reviewed the protocol. J.F.M., A.W., S.B. and B.P.N. cleaned and analysed the data. J.F.M., B.P.N., R.P., S.B., A.A.S.

and C.R.S. interpreted the data. J.F.M. wrote the first draft of the paper with input from J.P., B.P.N., P.M.A.A., A.A.S. and C.R.S. All authors read and commented critically on drafts of the paper. All authors approved the final version. C.R.S. supervised the entire work and is the guarantor.

## Competing interests

We declare that one or more of the authors have a competing interest as defined by Nature Portfolio. Professor Simpson reports grants from the UK National Institute for Health Research, the UK Medical Research Council, the NZ Health Research Council and the Ministry of Business Innovation and Employment. Dr Nguyen reports grants from the NZ Ministry of Business Innovation and Employment. Dr Janine Paynter reports grants (2021) and consulting fees (2019) from GlaxoSmithKline (all grants and consulting fees were institutional), the NZ Health Research Council, the NZ Ministry of Health and a travel grant from GAVI, The Vaccine Alliance. The remaining authors declare no competing interests.
