## [Peer Review File · Nature Communications]

Herpes zoster vaccine safety in the New Zealand population: a self-controlled case seriesREVIEWER COMMENTS

Reviewer #1 (Remarks to the Author):

Major comments:

The study applied a self-controlled case series method to evaluate several pre-specified serious adverse events that required hospitalizations following the zoster vaccine live vaccination which is an important contribution to the safety profile of ZVL.

The authors concluded no increased risk of serious AEs requiring hospitalizations following ZVL vaccination and compared to null findings (relative risk close to 1.0) from numerous previously published studies. However, they did not discuss the large (>50%) "protective" effects of the vaccine to most of the outcomes they studied (Table 2). These findings were not entirely 'similar' to the null findings from other studies.

Although the study included many potential negative controls, there were so few of these outcomes as shown in Table 3. I'm not sure how good these controls were in sufficiently addressing the potential bias in the study, given the low power to detect any differences with such small sample sizes. Are there other more common hospitalization outcomes that can serve as better negative controls?

The study sample included 1197 vaccinated people, but 1,130 pre-specified adverse events were identified after applying the outcome exclusion criteria. That means not all 1197 people had an AE in the "at risk" or the control periods. If that's true, the authors should report the number of people actually included (fewer than 1197) since only people with an AE during the "follow-up" period were included. The authors may also want to confirm that those vaccinated people without a qualifying AE (i.e. extra people included in 1197) were not included in their conditional models.

Minor comments:

Lines 91-93. The next paragraph said only 1,197 vaccinated people were included in the study population. So why were these 16081 Maori and 9170 Pacific people "meet the eligibility criteria and were included in the study"?

Lines 446-447. The "follow-up" period wasn't defined. I believe it included the "at-risk" period (1-42 days) AND the control period (72-162 days). It should be defined in this paragraph to set up the multiple use of "follow-up" period in the "Falsification outcomes" section.

Lines 447-448. "We were interested in who and when hence controlling for time-fixed confounding variables". I do not understand this sentence.

Line 460. It's important to note if outcomes occurred on day 0 (vaccination day) were included, which I don't think they should/were. In the abstract, at-risk window was defined as 0-42 days, implying outcomes on vaccination date were included.

Lines 464-465. "The post-risk period (post-vaccination self-comparison window) was 90 days, beginning on day 43 following vaccination." I believe the post-risk period began on day 72 (vs 43) following vaccination, since days 42-72 was a washout period?

Line 464. Why was the post-risk period set for 90 days instead of 42 days which was the length of the at-risk period?

Line 469. It should be "herpes zoster vaccination".

Line 508. Why did the authors use conditional logistic regression? Given essentially the same results from Poisson and logistic models in Table 2, I'd suggest to only use conditional Poisson.

Comments on Table 1.

- Should clarify "At risk" and "Control" in the columns
- Years should be "Vaccination year"
- Need footnote for NZDep2013 and NZDep2018. Instead of Quintiles 1-5, can the values of those quintiles be used instead?

Reviewer #2 (Remarks to the Author):

Thank you for the opportunity to review the paper, "Safety of the Herpes Zoster vaccine in the older New Zealand population: A self-controlled case series."

This is a well-designed observational study which adds to our understanding regarding the safety of live zoster vaccine. The study design is a self-controlled case series to assess pre-specified serious adverse events. Additionally, multiple negative controls were utilized to assess for confounding and bias. The conditional Poisson and conditional logistic regression analyses were appropriate given the study design and outcome.

The following recommendations are offered to improve the manuscript.

Discussion:

The authors need to explicitly state whether zoster live vaccine is still available for use in NZ. The live zoster vaccine was removed from the market in the US in 2020. The authors also need to state whether NZ guidelines now recommend inactivated/ recombinant zoster vaccine as the preferred, first-line vaccine to prevent herpes zoster. Availability and place-in-therapy for the live zoster vaccine in NZ are needed in the discussion in order for readers to put the study results in context. If the live vaccine has been withdrawn from the NZ market, the results are less compelling and less relevant to current practice.

Methods: Study Design:

Lines 464 and 465, correct the error in this sentence by replacing "43" with "73".

"The post-risk period (post-vaccination self-comparison window) was 90 days, beginning on day 43 following vaccination."

Methods: Prespecified Adverse Events:

In lines 480 and 481, please add more citations than just the one offered to support the following sentence. "The selection and classification of these serious adverse events were based on previous studies and biological plausibility"

I recommend acceptance of the paper with minor revisions.

Re: Nature Communications manuscript NCOMMS-23-14899-T: Safety of the Herpes Zoster Vaccine in the Older New Zealand Population: A Self-Controlled Case Series

REVIEWERS' COMMENTS

Reviewer #1:

Major comments:

The study applied a self-controlled case series method to evaluate several pre-specified serious adverse events that required hospitalisations following the zoster vaccine live vaccination which is an important contribution to the safety profile of ZVL.

Comment 1

The authors concluded no increased risk of serious AEs requiring hospitalisations following ZVL vaccination and compared to null findings (relative risk close to 1.0) from numerous previously published studies. However, they did not discuss the large (>50%) "protective" effects of the vaccine to most of the outcomes they studied (Table 2). These findings were not entirely 'similar' to the null findings from other studies.

Response

We have included the following in our discussion:

“Evidence from epidemiological studies have shown the risk of myocardial infarction and stroke, and transient ischaemic attack significantly increased in the first month following the reactivation of the varicella-zoster virus and decreased with time since infection. The risk of risk stroke is two to three times higher than the baseline level when reactivation occurs in the ophthalmic branch of the fifth cranial (trigeminal) nerve.²¹ A recent systematic review and meta-analysis and a nationwide retrospective matched observational study found that the zoster vaccine live was effective in preventing herpes zoster hospitalisation in people with cerebrovascular disease and heart disease.^{9, 10} In the current study, we identified a large (>50%) protective effect of the zoster vaccine live on most of the outcomes (stroke, acute myocardial infarction and heart failure) in the first 42 days following zoster vaccine live vaccination. A significantly reduced risk of stroke (relative incidence =0.58; 95% CI: 0.44, 0.78) following the zoster vaccine live immunisation has been reported in a recent self-controlled case series.¹⁶ The main difference between our findings and numerous antecedent studies is that the measure of association for the pre-specified adverse events was closer to one, or the confidence interval of the measure crossed one.^{5, 6, 8, 13, 16, 22} This variation may be due to the differences in study designs. The apparent protective effect observed in our study could be attributed to the zoster vaccine live effectiveness against herpes zoster and associated complications and healthy vaccinee time effect (At the time of vaccination and in the weeks immediately following, vaccine recipients exhibited a relatively healthier condition. However,

as time progressed, their health gradually reverted to the baseline level).^{14, 16} Although a previous study⁹ comparing the vaccinated (from which the current sample was selected) and the unvaccinated population found that both cohorts were comparable, further investigations are required to explore the significantly reduced risk of these outcomes following zoster vaccination. (Page 7, paragraph 1).

Response

Comment 2

Although the study included many potential negative controls, there were so few of these outcomes as shown in Table 3. I'm not sure how good these controls were in sufficiently addressing the potential bias in the study, given the low power to detect any differences with such small sample sizes. Are there other more common hospitalisation outcomes that can serve as better negative controls?

Response

We used fifteen negative controls. These include appendicitis, hernia, diverticulitis, femoral fracture, pertussis, cholelithiasis and cholecystitis, pancreatic diseases, sepsis, haemorrhoids, renal calculi, burns, lipoma, and epistaxis (see table below). Only diverticulitis, femoral fracture and cholecyst and pancreatic diseases are reported in Table 3, as we found no events (primary diagnosis) for the other indicators during the follow-up period. Our selection of negative controls was based on the following considerations:

- 1. We did not find any evidence in the literature to suggest that these conditions could be causally related to the zoster vaccine live.*
- 2. A recent self-controlled case series in Australia aimed at investigating the risk of pre-specified outcomes following live attenuated herpes zoster vaccine used only one negative control (burns). We equally used this negative control.*

Totterdell et al., (2020). Safety of live attenuated herpes zoster vaccine in adults 70–79 years: A self-controlled case series analysis using primary care data from Australia's MedicineInsight program. Vaccine. 38(23):3968-79.

- 3. Some of these indicators have been used to assess for bias in previous studies that compared the vaccinated and unvaccinated cohorts by vaccination status.*

Lzurieta et al., (2017). Effectiveness and Duration of Protection Provided by the Live-attenuated Herpes Zoster Vaccine in the Medicare Population Ages 65 Years and Older. Clinical Infectious Diseases. 64(6):785-93.

Tseng et al., (2011). Herpes Zoster Vaccine in Older Adults and the Risk of Subsequent Herpes Zoster Disease. JAMA. 2011;305(2):160–166. doi:10.1001/jama.2010.1983.

Tseng et al., (2012). Herpes Zoster Vaccine and the Incidence of Recurrent Herpes Zoster in an Immunocompetent Elderly Population, The Journal of Infectious Diseases, Volume 206, Issue 2, 15;190–196,

<https://doi.org/10.1093/infdis/jis334>

Jackson et al., (2006). Evidence of bias in estimates of influenza vaccine effectiveness in seniors. Int J Epidemiol. 35:337–44

Based on the above considerations, the indicators used are the most appropriate negative controls for this study.

We have added the following to the discussion:

"In the selection process for negative controls, a further consideration was made to try and identify controls which have a similar severity and prevalence to the pre-specified adverse events.²⁷ (Page 8; paragraph 1).

Also, we have added a table of all negative controls in the supplementary information.

Table S5: Falsification outcomes (page 9)

Outcomes	ICD-10-AM-iii Code
Appendicitis	K350; K351; K359; K36; K37; K380; K381; K382; K383; K388; K389
Hernia	K4000; K4001; K4010; K4011; K4020; K4021; K4030; K4031; K4040; K4041; K4090; K4091; K410; K411; K412; K413; K414; K419; K420; K421; K429; K430; K431; K439; K440; K441; K449; K450; K451; K458; K460; K461; K469
Diverticulitis	K5700; K5701; K5702; K5703; K5710; K5711; K5712; K513; K5720; K5721; K5722; K5723; K5730; K5731; K5732; K5733; K5740; K5741; K5742; K5743; K5780; K5781, K5782, K5783; K5790; K57921; K5792; K5793;
Femoral fracture	S7200; S7201; S7202; S7203; S7204; S7205; S7208; S7210; S7211; S722; S723; S7240; S7241; S7242; S7243; S7244; S727; S728; S729
Cholelithiasis and cholecystitis	K8000; K8001; K8010; K8011; K8020; K8021; K8030; K8031; K8040; K8041; K8050; K8051; K8080; K8081; K810; K811; K818; K819; K820; K821; K822; K823; K824; K828; K829; K830; K831; K832; K833; K834; K835; K835; K838; K839; K870
Pancreatic diseases	K85; K860; K861; K862; K863; K868; K869; K871; K903
pertussis	A370; A371; A378; A379
Sepsis	A40-A41
Haemorrhoids	I840-I849; O224; O872
Renal calculi	N20-N23
Burns	M6137-M6139; T200-T3199
lipoma	D170-D179
Epistaxis	R040

Comment 3

The study sample included 1197 vaccinated people, but 1,130 pre-specified adverse events were identified after applying the outcome exclusion criteria. That means not all 1197 people had an AE in the "at risk" or the control periods. If that's true, the authors should report the

number of people actually included (fewer than 1197) since only people with an AE during the "follow-up" period were included. The authors may also want to confirm that those vaccinated people without a qualifying AE (i.e. extra people included in 1197) were not included in their conditional models.

Response

We wish to confirm that those vaccinated people without a qualifying adverse event were not included in the conditional models. We have replaced Table 1 (Pages 12–14) to include only the number of people actually included in the conditional models after applying the outcome exclusion criteria. The replaced Table 1 has been moved to the supplementary material as Table S6 (Supplementary information, pages 10–12; Baseline characteristics of study participants before applying the outcome exclusion criteria) to complement Figure 2 (Data flow chart).

The baseline characteristics of study participants have been revised:

“A total of 1082 (0.4%) vaccinated people were hospitalised for a pre-specified adverse event of interest during the at-risk window (n=321) and control period (n=761) and were included in the study population. The mean age of the study population with an adverse event was 73.4 (standard deviation = 5.0; range = 58–92), and 809 (74.8%) were ≥ 70 years old. Furthermore, 59.3% (n=642) of these people were males, 72.0% (n=779) were European, and 8.6% (n=93) were immunocompromised (Table 1). Most of the study participants (24.5%) were from areas with the most deprived scores.” (Page 4, paragraph 2).

Table 1: Baseline characteristics of the study population

Characteristics	Total	At risk	Control
	N = 1082	N = 321 (29.67%)	N = 761 (70.33%)
Year of hospitalisation			
2018	658 (60.81%)	214 (66.67%)	444 (58.34%)
2019	237 (21.90%)	65 (20.25%)	172 (22.60%)
2020	117 (10.81%)	28 (8.72%)	89 (11.70%)
2021	70 (6.47%)	14 (4.36%)	56 (7.36%)
Age group			
50 – 64	≤ 5 (0.27%)	≤ 5 (0.31%)	≤ 5 (0.26%)
65 – 69	270 (24.95%)	80 (24.92%)	190 (24.97%)
70 – 74	321 (29.67%)	103 (32.09%)	218 (28.65%)
75 - 76	376 (34.75%)	109 (33.96%)	267 (35.09%)
≥ 80	112 (10.35%)	28 (8.72%)	84 (11.04%)

Sex

Male	642 (59.33%)	187 (58.26%)	455 (59.79%)
Female	440 (40.67%)	134 (41.74%)	306 (40.21%)

Ethnicity**(Level 1 ethnic codes)**

European (1)	779 (72.00%)	222 (69.16%)	557 (73.19%)
Māori (2)	147 (13.59%)	51 (15.89%)	96 (12.61%)
Pacific Peoples (3)	65 (6.01%)	18 (5.61%)	47 (6.18%)
Asian (4)	79 (7.30%)	25 (7.79%)	54 (7.10%)
Others (5,6,9)	12 (1.11%)	≤ 5 (1.55%)	7 (0.92%)

New Zealand Index of Deprivation 2013

Quintile 1	171 (15.80%)	47 (14.64%)	124 (16.29%)
Quintile 2	170 (15.71%)	45 (14.02%)	125 (16.43%)
Quintile 3	216 (19.96%)	69 (21.50%)	147 (19.32%)
Quintile 4	242 (22.37%)	73 (22.74%)	169 (22.21%)
Quintile 5	283 (26.16%)	87 (27.10%)	196 (25.76%)

New Zealand Index of Deprivation 2018

Quintile 1	162 (14.97%)	45 (14.02%)	117 (15.37%)
Quintile 2	191 (17.65%)	53 (16.51%)	138 (18.13%)
Quintile 3	225 (20.79%)	68 (21.18%)	157 (20.63%)
Quintile 4	239 (22.09%)	71 (22.12%)	168 (22.08%)
Quintile 5	265 (24.49%)	84 (26.17%)	181 (23.78%)

Immune suppression

Yes	93 (8.60%)	30 (9.35%)	63 (8.28%)
No	989 (91.40%)	291 (90.65%)	698 (91.72%)

District Health Board

Auckland	79 (7.30%)	26 (8.10%)	53 (6.96%)
Bay of Plenty	57 (5.27%)	17 (5.30%)	40 (5.26%)
Canterbury	129 (11.92%)	43 (13.40%)	86 (11.30%)
Capital & Coast	49 (4.53%)	12 (3.74%)	37 (4.86%)
Counties Manukau	116 (10.72%)	38 (11.84%)	78 (10.25%)

Hawke's Bay	34 (3.14%)	10 (3.12%)	24 (3.15%)
Hutt Valley	42 (3.88%)	12 (3.74%)	30 (3.94%)
Lakes	28 (2.59%)	≤ 5 (1.56%)	23 (3.02%)
MidCentral	58 (5.36%)	19 (5.92%)	39 (5.12%)
Nelson Marlborough	30 (2.77%)	9 (2.80%)	21 (2.76%)
Northland	57 (5.27%)	17 (5.30%)	40 (5.26%)
South Canterbury	16 (1.48%)	≤ 5 (0.62%)	14 (1.84%)
Southern	83 (7.67%)	23 (7.17%)	60 (7.88%)
Tairāwhiti	8 (0.74%)	0 (0.00%)	8 (1.05%)
Taranaki	17 (1.57%)	≤ 5 (1.56%)	12 (1.58%)
Waikato	109 (10.07%)	27 (8.41%)	82 (10.78%)
Wairarapa	11 (1.02%)	≤ 5 (0.62%)	9 (1.18%)
Waitemata	120 (11.09%)	42 (13.08%)	78 (10.25%)
West Coast	13 (1.20%)	6 (1.87%)	7 (0.92%)
Whanganui	26 (2.40%)	6 (1.87%)	20 (2.63%)

New Zealand Index of Deprivation (2013 and 2018) measures relative socioeconomic deprivation. It combines data related to communication, income, employment, qualifications, house ownership, support, living space and living condition. It is divided into quintiles (Quintile 1 = least deprived; Quintile 5 = most deprived), with each quintile representing 20% of the population.

At risk: Vaccine recipients with adverse events within 1 to 42 days post-vaccination.

Control: Vaccine recipients with adverse events within 73 to 162 days post-vaccination.

Minor comments:

Comment 4

Lines 91-93. The next paragraph said only 1,197 vaccinated people were included in the study population. So why were these 16081 Maori and 9170 Pacific people "meet the eligibility criteria and were included in the study"?

Response

The 16,081 Maori and 9,170 Pacific peoples are part of the initial vaccinated population of 278,375 adults.

This sentence has been rephrased to clarify the meaning:

"During the study period, 16,081 (5.8%) Māori and 9,170 (3.3%) Pacific peoples who received the zoster vaccine live." (Page 4, paragraph 1)

Comment 5

Lines 446-447. The "follow-up" period wasn't defined. I believe it included the "at-risk"

period (1-42 days) AND the control period (72-162 days). It should be defined in this paragraph to set up the multiple use of "follow-up" period in the "Falsification outcomes" section.

Response

We have included the definition for the follow-up period:

"The final study population comprised individuals with the outcome of interest that occurred during the follow-up (predefined observation) period. The follow-up period is the time after vaccination when we monitored the vaccinated cohort for any pre-specified adverse events and was divided into the at-risk (1–42 days) and the control (73–162 days) periods." (Page 22, paragraph 3)

Comment 6

Lines 447-448. "We were interested in who and when hence controlling for time-fixed confounding variables". I do not understand this sentence.

Response

We have changed the sentence to add clarity.

"This study design has the advantage of controlling for all time-fixed individual-level confounding variables as comparisons are within the same individual rather than between vaccinated and unvaccinated cohorts." (Page 22, paragraph 3)

Comment 7

Line 460. It's important to note if outcomes occurred on day 0 (vaccination day) were included, which I don't think they should/were. In the abstract, at-risk window was defined as 0-42 days, implying outcomes on vaccination date were included.

Response

This has been modified:

"The at-risk period was a duration of 42 days (days 1–42) following receipt of the zoster vaccine live vaccination (Figure 1). Pre-specified adverse events that occurred on the day of vaccination (index day; Day 0) were excluded from the analyses. The at-risk period of 42 days is consistent across pre- and post-licensure studies that assessed the safety of herpes zoster vaccines." (Page 22, paragraph 5)

Comment 8

Lines 464-465. "The post-risk period (post-vaccination self-comparison window) was 90 days, beginning on day 43 following vaccination." I believe the post-risk period began on day 72 (vs 43) following vaccination, since days 42-72 was a washout period?

Response

This has been modified:

"The post-risk period (post-vaccination self-comparison window) was 90 days, beginning on day 73 following vaccination" (Page 23, paragraph 1).

Comment 9

Line 464. Why was the post-risk period set for 90 days instead of 42 days which was the length of the at-risk period?

Response

We have included the following in the methods section:

The post-risk period was set at 90 days to increase the chance of capturing relevant adverse events. Also, 90 days of follow-up compensated for the absence of a pre-risk period. In a previous study conducted by Baxter et al. in 2012, the rate of all diagnoses of pre-specified adverse events of interest listed for hospitalisations and emergency department visits occurring in a 42-day risk period (days 1–42) following vaccination with herpes zoster live was compared to that in a 90-day postvaccination comparison period (days 91–180 post-vaccination), using vaccinees as their own controls.¹⁴ Also, the control period of 90 days has been used in recent studies to assess adverse events associated with live attenuated and trivalent inactivated influenza vaccination³⁶ and to evaluate the associations between exposure to ChAdOx1 or BNT162b2 vaccination and haematological and vascular outcomes using a nested incident-matched case-control study and a confirmatory self-controlled case series analysis.²⁵ (Page 23, paragraph 1).

Comment 10

Line 469. It should be "herpes zoster vaccination".

Response

This has been modified:

"Serious adverse events associated with herpes zoster vaccination were defined using the International Statistical Classification of Diseases and Related Health Problems, Tenth Revision, Australian Modification (ICD-10-AM-iii),³⁹ and extracted from the electronic medical records (NMDs)." (Page 23, paragraph 1)

Comment 11

Line 508. Why did the authors use conditional logistic regression? Given essentially the same results from Poisson and logistic models in Table 2, I'd suggest to only use conditional Poisson.

Response

Thank you for your suggestion. We would like to maintain both analyses. Our main analysis was conditional Poisson regression. We used conditional logistic regression as a sensitivity analysis.

Comment 12

Comments on Table 1.

- Should clarify "At risk" and "Control" in the columns
- Years should be "Vaccination year"
- Need footnote for NZDep2013 and NZDep2018. Instead of Quintiles 1-5, can the values of those quintiles be used instead?

Response

This has been modified (Reviewer 1, comment 3 above).

Reviewer #2:

Thank you for the opportunity to review the paper, "Safety of the Herpes Zoster vaccine in the older New Zealand population: A self-controlled case series."

This is a well-designed observational study which adds to our understanding regarding the safety of live zoster vaccine. The study design is a self-controlled case series to assess pre-specified serious adverse events. Additionally, multiple negative controls were utilised to assess for confounding and bias. The conditional Poisson and conditional logistic regression analyses were appropriate given the study design and outcome.

The following recommendations are offered to improve the manuscript.

Comment 1

Discussion:

The authors need to explicitly state whether zoster live vaccine is still available for use in NZ. The live zoster vaccine was removed from the market in the US in 2020. The authors also need to state whether NZ guidelines now recommend inactivated/ recombinant zoster vaccine as the preferred, first-line vaccine to prevent herpes zoster. Availability and place-in-therapy for the live zoster vaccine in NZ are needed in the discussion in order for readers to put the study results in context. If the live vaccine has been withdrawn from the NZ market, the results are less compelling and less relevant to current practice.

Response

We added this information to the discussion:

In New Zealand, the zoster vaccine live is approved for use in adults ≥ 50 years and used as part of the vaccination programme at age 65 from April 2018 to August 2022. The current guidelines preferentially recommend the recombinant zoster vaccine for the prevention of herpes zoster and postherpetic neuralgia in adults (≥ 50 years old) and immunocompromised people (≥ 18 years old). Two doses of the recombinant zoster vaccine are funded for adults 65 years old.³(Page 8, paragraph 4).

Comment 2

Methods: Study Design:

Lines 464 and 465, correct the error in this sentence by replacing "43" with "73".

"The post-risk period (post-vaccination self-comparison window) was 90 days, beginning on day 43 following vaccination."

Response

This has been modified:

"The post-risk period (post-vaccination self-comparison window) was 90 days, beginning on day 73 following vaccination" (Page 23, paragraph 1).

Comment 3

Methods: Prespecified Adverse Events:

In lines 480 and 481, please add more citations than just the one offered to support the following sentence. "The selection and classification of these serious adverse events were based on previous studies and biological plausibility"

Response

We have added more references as recommended:

The selection and classification of these serious adverse events were based on previous studies and biological plausibility.^{12-14, 16, 21} (Page 23, paragraph 2).

*R., Tran, T. N., Hansen, J., Emery, M., Fireman, B., Bartlett, J., . . . Saddier, P. (2012). Safety of Zostavax™—A cohort study in a managed care organisation. *Vaccine*, 30(47), 6636-6641. doi:10.1016/j.vaccine.2012.08.070*

*Simberkoff, M. S., Arbeit, R. D., Johnson, G. R., Oxman, M. N., Boardman, K. D., Williams, H. M., . . . Annunziato, P. W. (2010). Safety of herpes zoster vaccine in the shingles prevention study: a randomised trial. *Ann Intern Med*, 152(9), 545-554. doi:10.7326/0003-4819-152-9-201005040-00004*

*Totterdell, J., Phillips, A., Glover, C., Chidwick, K., Marsh, J., Snelling, T., & Macartney, K. (2020). Safety of live attenuated herpes zoster vaccine in adults 70–79 years: A self-controlled case series analysis using primary care data from Australia's MedicineInsight program. *Vaccine*, 38(23), 3968-3979. doi:https://doi.org/10.1016/j.vaccine.2020.03.054*

*Tseng, H. F., Liu, A., Sy, L., Marcy, S. M., Fireman, B., Weintraub, E., . . . Team, f. t. V. S. D. (2012). Safety of zoster vaccine in adults from a large managed-care cohort: a Vaccine Safety Datalink study. *Journal of Internal Medicine*, 271(5), 510-520. doi:10.1111/j.1365-2796.2011.02474.x*

*Wu, P. H., Chuang, Y. S., & Lin, Y. T. (2019). Does Herpes Zoster Increase the Risk of Stroke and Myocardial Infarction? *A Comprehensive Review. J Clin Med*, 8(4). doi:10.3390/jcm8040547*

Comment 4

I recommend acceptance of the paper with minor revisions.

Response

Thank you for recommending our paper

End

REVIEWERS' COMMENTS

Reviewer #1 (Remarks to the Author):

I thank the authors for responding to my questions. I have no further comment except for a note on a typo in Figure 2: there are 2 "Group 4" in the last rectangular box in the figure.

Re: Nature Communications manuscript NCOMMS-23-14899A: Safety of the Herpes Zoster Vaccine in the Older New Zealand Population: A Self-Controlled Case Series

REVIEWERS' COMMENTS

Reviewer #1:

Comment 1

I thank the authors for responding to my questions. I have no further comment except for a note on a typo in Figure 2: there are 2 "Group 4" in the last rectangular box in the figure.

Response

We have modified Figure 2 (page 22)

Figure 2: Data flow chart. A total of 278,375 adults were vaccinated during the study period (April 2018 to July 2021). A total of 1082 people developed pre-specified serious adverse events during the follow-up period and were included in the analysis. Stroke, Cerebrovascular diseases; Group 2: Cardiovascular events; Group 3: Meningitis, encephalitis, and encephalopathy; Group 4: Ramsay-Hunt Syndrome, Bell's Palsy; Group 5: Medically attended Reactions.

End